# Effectiveness of Vestibular Rehabilitation after Concussion: A Systematic Review of Randomised Controlled Trial

**DOI:** 10.3390/healthcare11010090

**Published:** 2022-12-28

**Authors:** Erasmo Galeno, Edoardo Pullano, Firas Mourad, Giovanni Galeoto, Francesco Frontani

**Affiliations:** 1Department of Scienze Mediche, Chirurgiche e Neuroscienze, Università degli Studi di Siena, 53100 Siena, Italy; 2Department of Clinical Science and Translation Medicine, University of Rome Tor Vergata, 00133 Rome, Italy; 3Departmental Faculty of Medicine and Surgery, Saint Camillus International University of Rome and Medical Sciences (UniCamillus), 00131 Rome, Italy; 4Department of Physiotherapy, LUNEX International University of Health, Exercise and Sports, 4671 Differdange, Luxembourg; 5Luxembourg Health & Sport Sciences Research Institute A.s.b.l., 50, Avenue du Parc des Sports, 4671 Differdange, Luxembourg; 6Department of Human Neurosciences, Sapienza University of Rome, 00185 Rome, Italy

**Keywords:** vestibular rehabilitation, traumatic brain injury, dizziness rehabilitation, vertigo rehabilitation

## Abstract

Introduction: Mild traumatic brain injury (mTBI) affects approximately 740 cases per 100,000 people. Impairments related to mTBI include vertigo, dizziness, balance, gait disorders double or blurry vision, and others. The efficacy on acute or chronic phase and dosage of vestibular rehabilitation (VR) in reducing these symptoms is not clearly stated. To clarify these points, we performed a systematic review of randomised controlled trials (RCTs). Methods: A systematic literature search was performed from 2015 to 2022 on PubMed, CINAHL, Cochrane Trial SPORTDiscus, Web of Science, and PEDRO. Eligibility criteria were RCTs which consider VR, participants with mTBI, and no gender or age restriction. Two blinded reviewers independently selected the study, and a third author was contacted in case of disagreements. Risk of bias was independently screened by two authors and successively checked by the other two authors. Results: Thirty-three full articles were read for potential inclusion and seven records met the inclusion criteria. The authors analysed different outcomes considering DHI, a meta-analysis was carried out, statistical difference was observed (*p* < 0.01), and a mean difference of −6.91 (−9.11, −4.72) in favour of VR was shown. Considering quality of life, the VR group reached a higher score on QOLIBRI. Controversial results were shown about balance and subjective symptoms questionnaire. Differently considering HiMAT, the authors showed a statistically important difference in favour of VR (*p* = 0.002). Conclusion: VR seems useful to reduce symptoms in patients with concussion; however, a huge heterogeneity of the studies and of the outcomes used were found. Therefore, a larger sample is necessary to assess the efficacy of VR.

## 1. Introduction

Discussion is ongoing of the proper management of mild traumatic brain injury (mTBI), more commonly termed concussion. There are approximately 740 cases of mTBI per 100,000 people, representing 55.9 million people each year. Sports-related traumatic brain injury (SR-TBI) is a very common occurrence. Each year, there are estimated to be from 1.6 to 3.8 million SR-TBIs, with an incidence that varies among the many types of sports [1]. Given this incidence, a proper dosage and effective conservative treatment seem important. Notably, mTBI is not a pathology that we find only in sports: other causes of mTBI are car accidents and accidental falls. Sensorimotor and vestibular impairments after concussion are well documented. They are classified as either peripheral or central and include vertigo, dizziness, balance and gait disorders, and double or blurry vision, among many others. It has been estimated that 30–65% of patients who have suffered a TBI will suffer vestibular symptoms (dizziness, nausea, vomiting, difficulty in concentrating, etc.). Impairments related to mTBI seem to be positively managed with vestibular rehabilitation (VR) [2,3], which consists in a set of treatments based on exercises which promote adaptation, substitution, habituation, and replacement (Table 1). The objectives of VR concern the improvement of gaze stabilisation, postural stability, symptoms of dizziness, and daily life activities [4]. However, it is necessary to compare this technique with other treatments and no treatment to determine its effectiveness. Murray et al. affirmed that VR’s apparent reduction of symptoms in patients after concussion is based on low-quality studies and that randomised controlled trials (RCTs) are lacking [5].

VR provides treatments oriented to the dysfunctional characteristics of patients (the ‘problem-oriented approach’) [5]. An in-depth evaluation is followed by the assignment of a personalised exercise programme, based on signs and symptoms related to the dysfunctions of other systems that may be involved as a result of the head trauma [6]. Considering the review of Murray et al. [5], the goal of our study was to analyse the effectiveness of the different techniques of VR, considering only high-quality studies such as RCTs to enable us to determine the effectiveness of VR compared to no treatment or other kinds of treatment. To reach our goal, a systematic literature review was conducted following the Preferred Reporting Items for Systematic Reviews and Meta-Analyses (PRISMA) guidelines [7]. The purpose of this systematic literature review was to analyse the VR dosage considered in the papers and what kind of sensorimotor or vestibular treatment seems most effective. Moreover, we hoped to differentiate the effectiveness of VR on patients with chronic, sub-acute, and acute symptoms. The results are summarised either as a meta-analysis or through a narrative approach.

## 2. Methods

### 2.1. Standards

The methodology of this review follows the Cochrane Handbook of Systematic Review [8], while the reporting adheres to the updated PRISMA Statement. Search reporting follows the guidelines for PRISMA literature searches reporting (PRISMA-S) [7,9].

The review was a priori registered in the PROSPERO database, registration number CRD42021247187 issued 5 May 2021. The inclusion criteria were that the papers should be full text, in the English language, include an RCT, and consider participants showing vestibular symptoms after concussion. 

Eligibility criteria: This study followed the participants, interventions, comparisons, outcomes, and study design (PICOS) framework.

Population: We considered adolescents and adults from 8 to 75 years old suffering vestibular symptoms related to a concussion following a trauma. No gender differences were considered.

Intervention: We considered all types of VR. VR is a specialised form of therapy intended to alleviate the impairments caused by vestibular disorders. Our review included rehabilitation programmes based on VR. 

Comparison: The following comparators were eligible: routine care, alternative care, pharmacological care, sham treatment, and wait and see. The most frequent treatments used as comparators were stretching and no intervention.

Outcomes: Outcomes considered were vertigo, dizziness, symptom modification, functional impact of vestibular symptoms, and balance. Outcomes were measured with specific questionnaires or tests related to the impairment or with general questionnaires related to patients’ quality of life. 

Study design: Only RCTs were included.

### 2.2. Search Strategy and Data Extraction (Information Sources)

The following databases were searched for studies published from 2015 to September 2022: PubMed, CINAHL, Cochrane Trial SPORTDiscus, Web of Science, and PEDRO. The search strategy was carefully designed by including vocabulary terms specific to each database and combined using the Boolean operators AND and OR (Appendix A). The following electronic databases were searched from inception to 1 September 2022: Medline, Web of Science, Cochrane Central Register of Controlled Trials (Cochrane CENTRAL), Cumulative Index to Nursing and Allied Health Literature (CINAHL), SPORTDiscus, and PEDRO. The grey literature was searched by one of the authors (EG). In addition, we screened all the studies included in the review performed by Murray et al. (2016) [5], and we added one paper which matched our inclusion/exclusion criteria. 

### 2.3. Study Selection

Titles and abstracts of the studies were screened by two blinded researchers (FF and EP) to identify eligible studies. The screening process was conducted on the rayyan.qcri platform. All the articles were evaluated and selected first by title, then by abstract, and, finally, by full text, according to the inclusion and exclusion criteria described previously. Abstracts deemed to have met the inclusion criteria by at least one reviewer were automatically retrieved as full-text articles. For those studies recommended for exclusion by at least one reviewer, a final decision was made by a third reviewer (EG), and any disagreements were arbitrated and assessed individually. Data extraction tables were created using Microsoft Word, and study design, population characteristics, outcome measures and follow-up intervals, interventions, results, and other relevant data were entered. Data that met the inclusion criteria were extracted by one person (EP) and independently verified by the other authors (FF, EG, FM, and GG). Extracted data were type of study (only RCTs), study characteristics (participants, age, time since concussion, and gender), and intervention (where information was available, prescribed exercise or prescribed VR was extracted). Treatment was described in terms of intensity (frequency, number of sessions, and duration of intervention), outcome measures (vestibular symptoms reported in a specified questionnaire, such as dizziness, gaze stabilisation, vertigo, gait impairments, return to sport (RTS), and quality of life), and results (the results related to the outcome were all reported). The outcomes were defined as short-term (three months), intermediate-term (six months), and long-term (one year). The corresponding author of every selected study was contacted via e-mail twice to check the results of their paper and receive additional information about it. 

### 2.4. Statistical Analysis and Narrative Synthesis

A meta-analysis of continuous or dichotomous outcomes was carried out whenever possible. The meta-analysis was performed using Review Manager (version 5.4.1, Cochrane Collaboration, Oxford, England), and the *p*-value was considered statistically significant at <0.05.

For outcomes where a meta-analysis was not possible, the result was presented as a narrative synthesis. The narrative approach was proposed considering all the outcomes of interest regarding post-concussion vertigo, dizziness, gaze stabilisation, and RTS reported in the selected papers. We decided to include data from the last follow-up of every paper. Where there were multiple follow-ups, we included data in which results were clearly different from the previous follow-up. We decided to exclude data which were missing, unclear, or unspecified. We included results that were scientifically admissible for every outcome regarding balance, dizziness, gaze stabilisation, quality of life, and RTS. 

### 2.5. Risk of Bias

The quality of the included studies was evaluated independently by two reviewers (EP and FF) using the Cochrane Risk of Bias Tool for RCTs. It was successively checked and accepted by the other reviewers (EG and GG). 

## 3. Results 

The systematic search retrieved 1492 records, with two additional articles retrieved from reference lists and the grey literature. Of these, 175 records were deleted with Endnote, 1126 records were deleted as unsuitable after the title was screened, and 162 were deleted after the abstract was screened. The remaining 31 articles were screened for full text. The study selection process is detailed in the PRISMA flow diagram (Figure 1). Twelve papers were excluded due to their study design, nine were excluded because of the treatment [10,11,12,13,14,15,16,17,18], and two were excluded because there was no clear distinction between mild, moderate, and severe traumatic brain injury [19,20]. Lastly, it was impossible to retrieve the full text of two papers, which were therefore excluded [21,22]. Six [23,24,25,26,27,28] records fulfilled the inclusion criteria. In addition, one study was included [29] from a previous review by Murray et al. (2016) [5] (Figure 1) (Table 2).

### 3.1. Patient Demographics

All the patients in the included studies experienced vestibular symptoms as the result of a concussion. The concussion occurred within 48 hours to 6 months before the baseline of the studies. A total of 7 RCTs were carried out and included a total of 266 participants (male = 100; female = 146). The gender of 20 participants was not specified [24]. The age of the included participants varied from 9 to 67 years. One study only considered an adolescent population [27], in two studies the sample considered was adolescent or adult [25,29], and the remaining four papers examined an adult population [23,24,26,28].

### 3.2. Types of Interventions

Considering the interventions, we analysed only those studies that considered VR as the main treatment. The comparators for VR were no treatment in three papers [23,26,29], sham or sub-therapeutic treatment in one paper [25], and stretching and physical activity in one study [27]; moreover, one study considered the same educational and physical activity programme for both groups plus VR only in the intervention group [28], and in one paper both groups underwent an identical drug treatment and one experimental group VR was added. [24].

### 3.3. Vestibular Rehabilitation

VR is described as an exercise-based treatment programme designed to promote vestibular adaptation and substitution. The goals of VR are (1) to enhance gaze stabilisation, (2) to enhance postural stability, (3) to improve vertigo, and (4) to improve activities of daily living. VR facilitates vestibular recovery mechanisms: vestibular adaptation, substitution by other eye-movement systems, substitution by vision, somatosensory cues, other postural strategies, and habituation (6).

Seven articles included in this review considered vestibular symptom reduction after a concussion. The protocol was similar in all seven papers, and the authors considered adaptation exercise, substitution exercise, habituation exercise, and balance/gait exercise to treat patients after mTBI. Four of these papers also considered manual techniques applied to the cervical spine [24,26,28,29], but only for patients who complained of cervical pain or range of motion restriction. The other papers considered a group VR with a tailored home-exercise programme. In three different papers [23,27,29], the control group did not receive any treatment. In one paper, both groups received the same medical therapy, but only one was given a VR protocol [24]. In two studies, patients received physical activity treatment [27,28], and in the last one, the control group received sham or sub-therapeutic treatment [25]. Of the studies included in this systematic review, only that by Schneider et al. [29] did not describe the exercise protocol properly. In the other papers, both progression and clinical reasoning for the proposed techniques were clearly stated [24,25,27,28,29].

### 3.4. Outcomes

We decided to summarise the results in terms of the outcome measures included in the studies. The data extracted were summarised in different sub-groups as follows. In regard to *balance,* the sub-groups are Activities-specific Balance Confidence Scale (ABCscale), Balance Error Scoring System (BESS), and modified BESS (mBESS). In regard to *dizziness*, they are Dizziness Handicap Inventory (DHI), Motion Sensitivity Quotient, and Vertigo Symptom Scale. *Quality of life* was investigated with QOLIBRI. *Subjective reports of post-concussion* were reported with the Rivermead Post-Concussion Symptom Questionnaire (RPQ), Post-Concussion Scale (PCS), and Post-Concussion Symptom Scale (PCSS). *Gait impairment* was investigated with functional gait assessment (FGA) and the High-level Mobility Assessment Tool for traumatic brain injury (HiMAT). *Gaze stabilisation and vestibular-ocular reflex (VOR)* were investigated with dynamic visual acuity (DVA), the head impulse test (HIT), and the Vestibular/Ocular Motor Screening (VOMS) scale. Other outcome measures were used at the baseline to assess the VOR and other reflexes related to vestibular symptoms, such as the head thrust test, but this measure was not further discussed because it was not analysed as an outcome measure by the authors. The last outcome analysed was *return to sport (RTS)*, defined as medical clearance to return to full sports activity. Heterogeneity of outcome measure and follow-up allowed a meta-analysis only for the DHI score considering short-term outcomes. Adverse responses were not documented.

### 3.5. Risk of Bias

Only one paper showed a high risk of bias [24]. The other papers could be defined as at low risk of bias. All papers except [24] used an adequate sequence generation and allocation procedure, so in six studies there was a low risk of selection bias. Unfortunately, none of the papers showed blinding of personnel or participants, so all papers had a high risk of performance bias. Reporting bias was shown by Schneider et al. and Jafarzadeh et al. [24,29]. No papers except [24] showed attrition bias, and all except [24] gave an adequate overview of withdrawal or drop-outs (Figure 2).

### 3.6. Efficacy of Intervention: Analysis (Synthesis of Results)

#### 3.6.1. Return to Sport

RTS criteria were analysed in two papers. Schneider et al. [29] found a *p* < 0.001 for medical clearance to RTS in the treatment group within eight weeks of treatment; thus, a ratio of 3.91 times more individuals in the treatment group were medically cleared to RTS than in the control group. Reneker et al. [25] showed a rate of 2.91 in favour of the experimental group in the time-to-medical release RTS, with a median number of days to medical release of 15.5 for the experimental group and 26 for the control group [25,29].

#### 3.6.2. Dizziness

Dizziness was analysed using the DHI scale in five papers [23,24,27,28,29], the vertigo symptoms scale in one paper [23], and Motion Sensitivity Quotient in one study [29]. A meta-analysis of the DHI scale and data used by Kleffelgaard et al., Jafarzadeh et al., and Kostos et al. was carried out. The data showed a mean difference score of −6.91 (−9.11, −4.72) in favour of the experimental group with a statistically significant difference between groups (*p* < 0.001) (Figure 3). Quantitative analysis of the results reported a statistically significant difference between group mean in DHI at the first follow-up (*p* = 0.03) but no difference at the second follow-up (*p* = 0.09) [23]. The mean difference between groups was −8.7 (−16.6 to −0.9) in favour of the intervention group [23]. Jafarzadeh et al. [24] showed a significant difference in weeks 3 and 4 in favour of the VR group (*p* < 0.001) with a mean change in DHI between the groups of 20 ± 11 in the intervention group and 0.2 ± 7.8 in the control group at four weeks, during the last follow-up. Schneider et al. [29] reported median DHI changes in the score of −24 in the intervention group and –48 in the control group (just one participant) in those cleared to RTS. Moreover, the authors found a median change of −13 in the intervention group and –21 in the control group in those not cleared to RTS. Kostos et al. [27] reported a small treatment effect for the overall score of the DHI with a mean score of −24.94 (SD 3.72) on the intervention group and a mean score of −17.89 (SD 3.61) on the control group, with no statistically significant difference between groups (*p* = 0.18). Langevin et al. [28] showed no statistical difference between group interaction but only by time interaction for both groups. The DHI score decreased from 45.8 (SD 22.1) in the experimental group and 44.8 (SD 21.87) in the control group to 11.86 (SD 12.16) in the experimental group and 9.63 (SD 9,32) in the control group, respectively. The other dizziness outcome measure employed by Schneider et al. [29] was the modified Motion Sensitivity Quotient. In the treatment group, median changes of −10 and −1.75 were reported in those cleared and not cleared, respectively (to RTS). Conversely, in the control group, median changes of −20 and −7.25 were observed in those cleared and not cleared, respectively (to RTS). Lastly, in regard to dizziness, Kleffelgaard et al. [23] were not able to detect statistically significant changes in the Vertigo Symptom Scale with a mean difference of −2.1 (−4.5 to –0.2) (*p* = 0.08) in the vertigo subscale and 0.4 (−1.4 to −2.1) (*p* = 0.69) in the anxiety subscale. 

#### 3.6.3. Subjective Reports of Concussion Symptoms

This outcome measure was reported by PCS, Rivermead Post-Concussion Symptoms Questionnaire, and PCSS. The PCS was used by Reneker et al. [25], who reported that, when accounting for a history of previous concussion, the experimental group recovered at a rate of 1.99 compared to the control group (HR: 1.99; 95% CI: 0.95, 4.15). Kleffelgaard et al. [23] reported no statistically significant difference at either follow-up in the Rivermead Post-Concussion Symptoms Questionnaire. The analysis of the PCSS performed by Kostos et al. [27] showed no statistically significant difference between the group and a small effect in favour of the treatment. PCSS analysis performed by Langevin et al. [28] showed no group by time interaction statistically significant difference between groups for the total score (*p* = 0.62) but a time effect with a significant improvement for both groups. The PCSS score decreased from 62.83 (SD 23.69) in the experimental group and 61.77 (SD 22.59) in the control group to 13.96 (SD 12.63) in the experimental group and 14.67 (SD 16.85) in the control group at 26 weeks post-baseline [28].

#### 3.6.4. Balance

Considering balance impairment, the BESS, used by Kleffelgaard et al. [23], showed no statistically significant difference between groups (*p* = 0.15). Considering balance impairment, Schneider et al. [29] reported that 64% of those medically cleared to RTS in the treatment group reached the maximum score of 100/100 on the ABCscale compared with 25% of the control group who were not medically cleared. Considering the mBESS, no difference was found by Kostos et al. [27] with a small effect size in favour of the treatment. 

#### 3.6.5. Gait Impairment

Gait impairment and dynamic balance were considered by Schneider et al. and Kleffelgaard et al. [23]. The first author group showed an FGA improvement in the intervention group of 1 in those cleared to RTS and of 3 in those not cleared to RTS. In contrast, in the control group, there was a median FGA change of 3 in those cleared to RTS and of 1 in those not cleared to RTS. Kleffelgaard et al. [23] showed a statistically significant difference at the first follow-up considering the HiMAT with a between group mean difference (*p* = 0.002); no difference was shown at the second follow-up (*p* = 0.09). 

#### 3.6.6. Quality of Life

Quality of life was analysed by only one author group, Soberg et al. [26]. Considering the QOLIBRI, the multivariate model showed a mean of 6.5 points higher in the intervention group than in the control group for the change scores on the QOLIBRI. 

#### 3.6.7. Gaze Stabilisation

In regard to gaze stabilisation and VOR, Schneider et al. [28] did not perform a specific test between groups of the DVA so the results could not be analysed. The VOMS was analysed by Kostos et al. [27] and showed that a statistically significant difference existed between horizontal VOR (*p* = 0.04) and vertical VOR (*p* = 0.01) but not visual motion sensitivity (*p* = 0.07). However, a medium to large treatment effect was shown in all the reflexes. A statistically significant difference was also shown in the horizontal and vertical VOR by Langevin et al. [28], who showed a group by time interaction difference in favour of the VR group at six weeks only (*p* < 0.01). Moreover, the authors showed a group by time interaction was reported for the HIT (*p* < 0.01) at 12 weeks. For the other reflexes analysed, no differences were reported.

## 4. Discussion

To the best of our knowledge, this is the first systematic review to analyse only RCTs considering VR following concussion. As a starting point, we used the review performed by Murray et al. in 2016 [5]. The use of VR is an emerging topic in rehabilitation. In fact, many existing studies suggest the usefulness of VR in patients with mTBI/concussion who experience persistent vertigo or balance symptoms. The definition of persistent symptoms varies in the literature from seven days to three months. Hence, a huge variety of patients is included in the studies in terms of impairments, and our review reflects this heterogeneity. After TBI, patients are diagnosed within the first few days or even weeks after the traumatic event; however, the optimal time to begin VR following injury remains unclear [5]. Hence, one of the goals of our review was to differentiate between acute, persistent, and chronic cases and to determine potential differences in dosage and treatment over the time which elapses after concussion. Unfortunately, we are not able to clarify this point. In fact, we found that three papers included only acute patients [25,27,29], one paper included a patient with persistent symptoms between 3 and 12 weeks after the concussion [28], and the other three studies did not clearly state the time course from the concussion [23,24,26] and the inclusion on the study, making it difficult to compare the results of these papers considering the differences in time elapsed since concussion. 

### 4.1. Acuity and Return to Sport

The two papers [25,29] which involved acute patients indicate that VR is effective in reducing the time to clearance to RTS. In fact, both papers showed a reduced time to RTS. The paper by Reneker et al. [25] is really interesting: it compares VR to sham and sub-therapeutic treatment, showing that the treatment has an important impact and that there is a significant interaction between treatment and positive history of concussion. The number of sessions was the same in both studies, which considered a maximum of eight treatments, but the frequency was different: one paper considered two treatments per week [25], and the other considered one treatment per week [29]. However, huge differences exist in regard to the RTS definition, so these results should be considered carefully. In the other papers included in this review, the authors did not take proper account of the time elapsed since concussion and the beginning of the treatment, and many different outcomes were used to analyse the effectiveness of VR. Below, we discuss the differences in the analyses of outcome measures.

### 4.2. Dizziness

Considering the DHI, it was possible to perform a meta-analysis of three studies [23,24,27] which showed a significantly statistical difference in favour of the intervention (*p* < 0.001). The mean difference between groups was −6.91 (−9.33, 4.77), and it has recently been suggested that the minimum important difference between groups is 6 [30]. This difference could be considered important not only statistically, but also clinically. In two of the three studies which considered the DHI, the authors found that a statistical difference was present between the intervention and control groups. Unfortunately, the paper by Jafarzadeh et al. [24] had a high risk of bias. Langevin et al.’s [28] analysis did not show any statistical difference between groups for the DHI. In this study [28], the authors considered gradual physical activity for both groups of treatment. Gradual physical activity is considered a fundamental option for the reduction of the symptoms after concussion. So, adding VR to gradual physical activity does not seem to modify significantly the DHI score. In fact, for both groups, a time interaction improvement was demonstrated. Additionally, Schneider and al. [29] showed a difference in the median score of DHI for participants who were cleared to return to sport. Considering the other dizziness outcomes measures used by the authors, it was not possible to collect any differences between groups. So, VR seems effective to reduce symptoms reported on DHI but not so effective when considering Vertigo Symptom Scale. 

### 4.3. Subjective Reports of Concussion Symptoms

Four authors analysed different outcome measures on this topic [23,25,27,28]. Reneker et al. [25] showed a favourable VOR in the VR group, taking into account a previous history of concussion. Kleffelgaard et al. [23], Kostos et al. [27], and Langevin et al. [28] did not show any statistically significant differences between groups. Thus, the subjective reports did not seem to follow the improvement shown in dizziness, VOR, and gait impairments but seemed more related to balance, which did not improve significantly in the studies analysed in this paper.

### 4.4. Balance

Kleffelgaard et al. [23] did not find any differences between intervention and the control group considering balance outcomes. Likewise, Kontos et al. [27], who considered the mBESS, did not find any differences between groups. As only two papers considered balance outcomes which could be really important for patients with vestibular symptoms, in the future it could be interesting to focus on this symptom. 

### 4.5. Gait Impairments

Schneider et al. and Kleffelgaard et al. [23,29] considered gait impairment as an outcome of VR following a concussion. The authors showed that, in the short term, VR could reduce the gait impairment of patients. The authors considered the same dosage over eight weeks of treatment for each group, but the number of sessions was different: Kleffelgaard et al. [23] considered two treatments per week while Schneider et al. [29] considered one session per week. Schneider et al. [29] also analysed the gaze stabilisation which could affect patients after mTBI, but the results were not properly analysed so it is not possible to describe them. 

### 4.6. Quality of Life

Lastly, in patients who were affected by concussion, considering HRQL on the QOLIBRI, a significant effect in favour of the VR was shown with a mean score 6.5 points higher compared to the control group [26]. 

### 4.7. Gaze Stabilisation and VOR

Three studies considered VOR and gaze stabilisation as outcome measures and showed contradictory results. In fact, Schneider et al. [29] demonstrated no differences in this outcome measure, whereas, in relation to VOR, Kontos et al. [27] and Langevin et al. [28] showed a statistical difference for vertical VOR and horizontal VOR. Hence, these results need further investigation if we are to understand whether VR could be helpful in the short term in reducing VOR reflexes. The HIT analysis performed by Langevin et al. [28] showed a group by time interaction in favour of the VR (*p* < 0.01) at 12 weeks after the baseline. 

From current evidence, it therefore appears that VRT is a valid treatment strategy for the management of patients suffering from unilateral vestibular peripheral problems [31]. Considering our results, we could affirm that, in the acute phase, VR seems effective in reducing time to RTS. Moreover, a meta-analysis showed that dizziness seems to be positively influenced by VR in the short term. These results were not confirmed by a qualitative analysis over the long term. Moreover, gait impairment seems to be positively affected by VR in the short term. All these beneficial effects of VR could improve the quality of life of patients, as was shown by Soberg et al. [26], but unfortunately only one paper considers this impairment. However, patients showed no difference in the subjective reports of concussion symptoms, although improvement in quality of life and different outcome measures was shown in almost all the studies included. We consider the prescription and dosage of VR for patients with TBI to be a very important issue, but this was largely addressed in a previous review [5], and we were unable to retrieve any new information on this topic. Clinically, we very often observe a delay in the prescription of VR treatments, and this could pose a problem considering the results of our review. In fact, the effectiveness in the acute phase shown by Reneker et al. [25] and Schneider et al. [29] seems relevant. Considering the papers included in this review proposed a huge range of dosage treatments to patients, we cannot suggest an optimal dosage for administering specific exercises in this category of patients. These factors, together with the disproportionate use of vestibular suppressor medications [32], can certainly affect the prognosis and chronicity of symptoms, as has already been demonstrated in other clinical presentations [33]. Unfortunately, the effectiveness of VR on chronic symptoms is not clear because none of the studies included in the current work only considers patients with chronic symptoms. However, in acute or persistent conditions, VR seems useful in reducing some symptoms related to concussion. Finally, we can affirm that the management of a patient suffering from sensorimotor system disorders after a TBI should include a physiotherapy evaluation. The goal of this assessment is to identify physical, biological, and psychological impairments, as well as functional disabilities, and evaluate, together with the physicians, when to start a rehabilitation programme. Unfortunately, further studies are needed to obtain details concerning the initiation of VR treatment, the dosage, the time course from concussion to the beginning of the treatment, and further RCTs to reinforce the conclusions of our study.

## 5. Limitations

The first limitation was the heterogeneity of the studies, considering the different periods of time which had passed since concussion (the range was 48 hours to 6 months); moreover, due to the different outcomes and follow-up used by the authors, it was not possible to properly compare the efficacy of vestibular treatment. One study did not accurately describe the VR; however, the other studies clearly defined the progression of the exercises prescribed. A larger sample and better codification of the time between concussion and treatment are necessary. The VOR and other vestibular interaction reflexes (OKN, VRS, and others) were not analysed in all the studies, so it could be interesting to add them as outcome measures in future studies.

## 6. Conclusions

VR seems to be a valid approach for the management of patients suffering from dizziness after concussive trauma. VR seems to reduce the time to clearance to RTS in the acute phase and to modify quality of life and gait impairment symptoms in patients who have suffered an mTBI. Moreover, a meta-analysis showed that DHI scores improved significantly in the short term (*p* < 0.01). Considering this relevant outcome, VR could be a valid approach in the short term. We need more studies with higher magnitude and that properly consider the time elapsed since concussion to detect the correct approaches and dosage.

## Figures and Tables

**Figure 1 healthcare-11-00090-f001:**
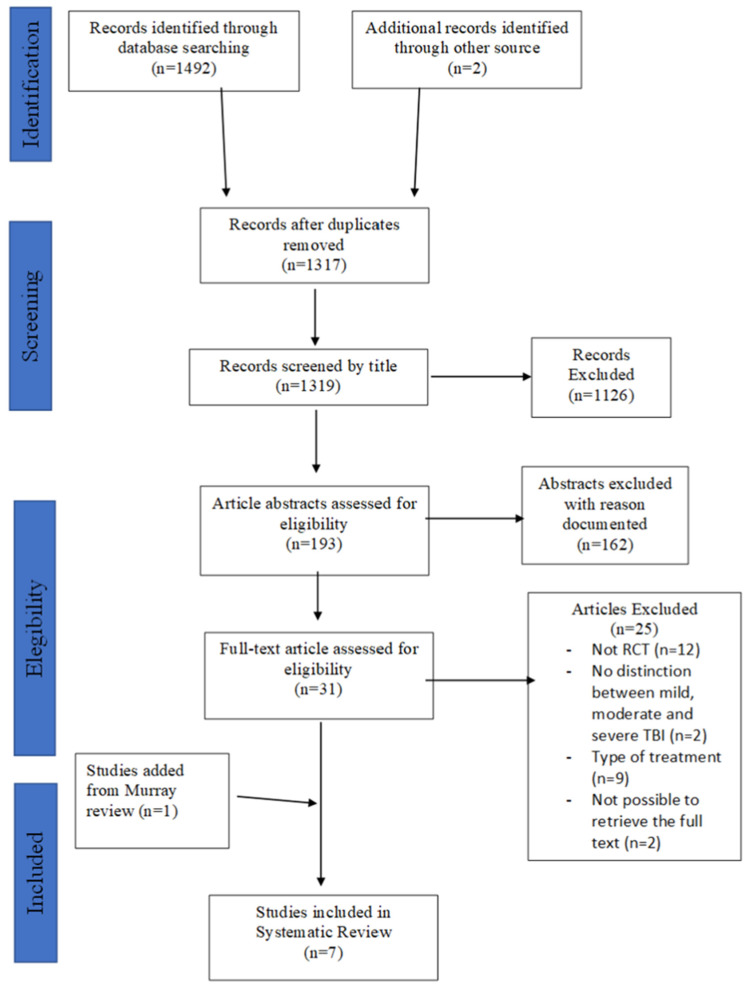
PRISMA flow diagram for systematic search. TBI, traumatic brain injury; VRT, vestibular rehabilitation therapy.

**Figure 2 healthcare-11-00090-f002:**
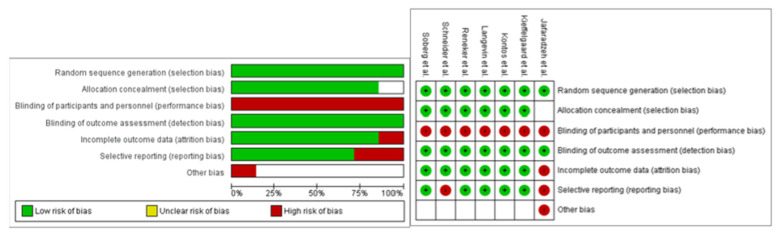
Tables representing the risk of bias for the included studies.

**Figure 3 healthcare-11-00090-f003:**
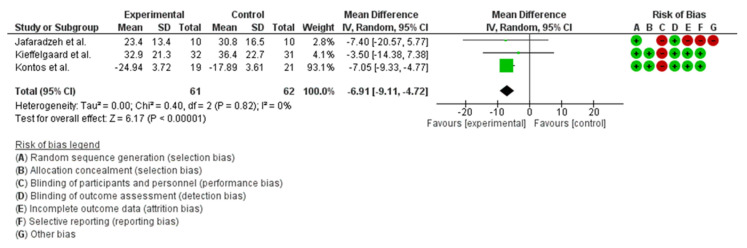
Figure representing the results for the meta-analysis for DHI outcome.

**Table 1 healthcare-11-00090-t001:** Brief description of the different vestibular rehabilitation techniques.

**Adaptation Exercise**	process where nerve impulses in the brain are able to shift or “adapt” to the incorrect signals from the damaged vestibular system. This gradual shift allows your brain to recalibrate itself.
**Substitution**	recovery principle uses other body functions or strategies to replace the missing vestibular function.
**Habituation**	process allows you to gradually desensitize yourself to vestibular movement and stimulation if you are repeatedly exposed to it.
**Replacement**	different repositioning maneuvers can be performed to help resolve the spinning that occurs due to position changes.

**Table 2 healthcare-11-00090-t002:** Summary table of main characteristic of the studies.

Study Author/Year	Type of Study	Sample	Intervention	Outcome Measure	Results
Soberg H.L. et al. (2021) [26]	Single blind RCT	n = 64 (19 males, 45 females). Mean age was 39.4 (SD 13.0). There was a measure at the baseline (T0) then at the first follow-up (T1) at 2.7 (SD 0.8) months after the baseline. The second follow-up (T2) was 4.4 (SD 1.0) months after the baseline.	Both groups received the TAU Intervention Group: TAU combined with an individualised group-based VR programme, 16 sessions in 8 weeks. VR exercises were tailored and described in another study (27). Control group: only TAU.	QOLIBRI and HRQL were the main outcome measures. RPQ, VSS-SF, and HADS are the secondary outcome measures.	Significant group effect in favour of the intervention group in HRQL on the QOLIBRI. The score at T0 of the QOLIBRI was between 45.4 and 66.7 (SD between 19.2 and 22.7), while at T2, the score was between 55.3 and 66.6 (SD between 20.3 and 24.7). The *p*-value for the QOLIBRI was <0.02.
Reneker J.C. et al. (2017) [25]	Double-blind RCT	n = 41. The population included athletes, participating in sports aged 10–23 years with an acute concussion and dizziness diagnosed with PCS. The intervention group (n = 22) with a mean age of 16.5, control group (n = 19) with a mean age of 15.9. The follow-up was made after a 4-week period.	Group 1: The PT designed an individualised and progressive treatment plan. VR included different techniques (including habituation and adaptation), oculomotor control, neuromotor control (including proprioceptive and kinesthetic awareness), and balance exercises were added to each subject’s treatment regimen as indicated Generally, each intervention session lasted between 30 and 60 minutes. Group 2: The PT delivered interventions that ranged from sham, sub-therapeutic, and non-progressive therapeutic techniques to minimally progressive therapeutic techniques.	Primary outcomes: symptomatic recovery with PCS and medical clearance for RTP.	The median time for medical release was 10.5 days sooner in the experimental group than in the control group. The median time for PCS recovery was 3.5 days sooner in the experimental group than in the control group. Considering Cox proportional hazards regression for time until medical release for RTP, the experimental group demonstrated a hazard ratio of 2.91 compared to the control group. (95% CI: 1.01, 8.43).
Jafarzadeh, S, et al. (2018) [24]	RCT	n = 20 adult patients (aged 18–60 years). Patients had a mean age of 44.2 (SD 12.6). The follow-up was after 4 weeks of rehabilitation.	Participants were randomly divided into two groups. Control Group: received the usual medical therapy (Betaserc 8 mg pills; at least three pills per day). Intervention Group: received medical therapy and VR after a 4-week period. Different VR techniques were proposed considering the baseline condition of the patients. Different gaze stabilisation and adaptation exercises were used in all patients, although substitution exercises including standing and walking exercise were used only in patients with unsteadiness. More detailed data were summarized in the study.	DHI	Early vestibular rehabilitation programme can decrease vertigo symptoms and increase stability and balance performance. Medical therapy group at week one was 1.8 (SD = 10.9) while at week four was 0.2 (SD = 7.8). The medical therapy and vestibular rehabilitation group at week 1 was −2.0 (SD = 8.7) while at week four was 20.0 (SD = 11.0) with *p* = 0.000.
Kleffelgaard I. et al. (2019) [23]	RCT	n = 65 with TBI (45 females and 19 males). Intervention group (n = 32) with a mean age of 37.6 (SD 12.3) and control group (n = 31) with a mean age of 41.2 (SD 13.6). Baseline at 3.5 (mean) months after injury. First follow-up at a mean of 2.7 months. Second follow-up at two months after the end of the intervention.	Control group: (n = 32) did not receive any rehabilitation intervention. Intervention group: (n = 33) received a group-based vestibular rehabilitation. VR exercises were tailored and described in another study (27). The intervention was twice weekly for eight weeks. Both groups received usually multidisciplinary outpatient care.	Primary outcome: DHI Secondary Outcome: HiMAT, VSSV, VSSa, RP3, RPQ13, HADSa, HADSd, and BESS	First follow-up, statistically significant mean differences in favour of the intervention were found in DHI (−8.7 points, 95% CI: –16.6 to −0.9) and HiMAT (3.7 points, 95% CI: 1.4–6.0). The *p*-value was significant for first follow-up: the DHI *p* = 0.03 and the HiMAT *p* = 0.002. No significant difference in other outcomes.
Schneider M.J. et al. (2014) [29]	RCT	Treatment group (n = 15): 11 males, 4 females. Median age: 15 (SD 12–27). Control group (n = 16): 7 males, 9 females. Median age: 15 (SD 13–30).	Both groups performed non-provocative range of motion exercises, stretching, and postural education. Treatment group: in addition, received an individual designed vestibular rehabilitation and cervical spine physiotherapy. VR includes an individualised programme of habituation, gaze stabilisation, adaptation exercises, standing balance exercises, dynamic balance exercises, and canalith repositioning manoeuvres.	(1) Number of days until medical clearance to return to sport. (2) 11-point Numeric Pain Rating Scale score, ABC scale, DHI, SCAT2, DVA, head thrust test, modified motion sensitivity test, FGA, CFE, and JPE.	Return to Sport: OR 10.27, *p* < 0.001 for return to sport in 8 weeks for the intervention group. Intention to treat analysis: OR 3.91 (95% CI 1.34 to 11.34) for the treatment group to be medically cleared to return to sport compared with the control group, (*p* = 0.002). No between-group analyses for secondary outcomes were reported.
Kontos A.P. et al. (2021) [27]	RCT	Treatment group (n = 25): 16 females, 9 males. Median age: 15.3 (SD 1.6). Control group (n = 25): 15 females, 10 males. Median age: 15.3 (SD 1.7). The outcomes were recorded at 2 and 4 weeks post-intervention. The participants who were recovered by 2 or 4 weeks stopped the intervention and completed the clinical outcomes.	Both groups performed a behavioural management. Treatment group performed also individual VR and home VR exercises for 30 minutes per day. Control group: performed stretching and physical activity for 30 minutes per day.	VOMS: to assess the VOR, DHI, mBESS, and PCSS	There was a medium treatment effect size for horizontal VOR and VMS (0.09–0.11) and large for vertical VOR (0.16). The subscales of DHI-F demonstrated a medium treatment effect size (0.06–0.1), whereas all other secondary outcomes demonstrated a small treatment effect (0.01–0.06). Significant statistical difference was shown only for horizontal VOR (*p* = 0.04) and vertical VOR (*p* = 0.01). No other significantly differences were shown.
Langevin P. et al. (2022) [28]	RCT	Treatment group: (n = 30): 20 females, 10 males. Mean age: 38.9 (SD 14.56) Control group (n = 30): 21 females, 9 males. Mean age: 39.07 (SD 12.63). The outcomes were recorded at baseline, and after 3, 6, 12, and 26 weeks.	Both groups received education and advice about exercise tolerance and concussion. Control group received 8 sessions in 6 weeks of supervised cardiovascular exercise. Treatment group received the same treatment as control group +1 to 8 sessions of cervicovestibular treatment. The treatment consisted in manual and therapeutic exercises for cervical spine and repositioning manoeuvre, vestibular adaptation, ocular motor exercise, balance, and habituation exercise	PCSS, DHI, NPRS, clearance to return to function, VOMS, and head impulse test (HIT).	No group by time interaction difference was observed for PCSS, DHI, NPRS, and return to function. All the groups demonstrated a statistically significant difference from the baseline. A group by time interaction was observed for horizontal and vertical VOR in favour of the treatment group at 6 weeks (*p* < 0.01). A difference for group interactions was observed for HIT (*p* < 0.01).

## Data Availability

All the data generated or analyzed during this study are included in this published article.

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
