# Peer review of "Effectiveness of Vestibular Rehabilitation after Concussion: A Systematic Review of Randomised Controlled Trial"

_healthcare, 2022, doi:10.3390/healthcare11010090_

Round 1

Reviewer 1 Report (Previous Reviewer 1)

Thank you for making the suggested revisions. This is a much cleaner paper that is very focused and rigorously reviews the available (paucity) of RCTs on this subject. The text is well organized and flows in a more logical manner. Thank you also for keeping this focused on "traditional" vestibular rehabilitation and removing the study on gaming that was incongruous. 

Reviewer 2 Report (Previous Reviewer 2)

The changes improved the quality of the paper

I've no other remarks

This manuscript is a resubmission of an earlier submission. The following is a list of the peer review reports and author responses from that submission.

Round 1

Reviewer 1 Report

Thank you for putting together this very interesting systematic review. I feel your overall paper was well written and based on sound principles of analysis. After your rigorous search of the literature there is an obvious paucity of RCTs studying the effectiveness of VR in patients with mTBI.  There are two fundamental changes that will make this a much cleaner paper: 1.) Remove the study and report on the RCT using the video gaming system. Although it is interesting in its own right, it falls outside of the A-B (VR-Control) review design. By keeping it in, you have an A-B-C (VR-Control-Gaming) review design and would need more studies on video gaming for this design to make sense. In addition, the analysis of A-B-C designs are more complex and would require more sophisticated statistical analysis. 2.) I would remove the "return to sport" outcome measure in the abstract (but keep your findings in the body of the work). It is one of many outcome measures discussed in the paper Having it in the abstract overemphasizes its importance and sets up the expectation that this is a review of athletes with concussion and the effectiveness on VR with return to sport.  Your population is clearly broader than this (which I am pleased to see) and you don't want to minimize this. Readers might pass on reading further because sport is not in the realm of their practice and would miss out on the important and relevant information you have presented. 

In terms of overall organization of the paper, you do a wonderful job in breaking-out the Introduction, Methods, and Analysis including subsections. This keeps it very organized and readable. I would recommend carrying that through in your final two sections. Starting at line 286, it will be much more readable if you organize the outcome data for VR by subsections (DHI, etc.) This will allow the reader to jump to a particular outcome measure and take a deeper dive into the results. You have done this nicely in table 2 and mirroring this structure in the text above would help the density of the information to be more digestible. I would make the same recommendation for your discussion beginning on line 342. At the very least, organize each point to be discussed as their own paragraph (topic sentence and supporting examples), but delineating sub-sections (acuity, outcomes, etc.) will help make your important points standout and this section more readable. 

Again, thank you for sharing the results of your work. I am sure that others will find your insights valuable and inspire others to do more RCTs with this population!

Author Response

We are thankful for your consideration and suggestion about our study. We tried to improve our work following your indication. In the following part you will find the point-by-point reply.

As suggested, we removed from the abstract the consideration about Return To Play (RTP) and Video Game Therapy (VGT) (line 25-26 and 28). We modified the abstract adding information about the other outcome measures analysed in our systematic review (line 29-31).

As suggested, we removed from the manuscript the section which consider the VGT and all the reference in the different parts of the text (line 85, 131-133, 167-169, 190-192, 212-216, 275,287-291,305, 331-335,  352-353).

As suggested, we modified the final two sections, as you can see we added different subsections which underline the outcome measure for the vestibular rehabilitation section and discussion. In the discussion we underline also the acuity as an important subsection together with the RTP because it was impossible for us to separate the findings about this topic.

Reviewer 2 Report

The paper is well written and selection of works clearly explained.

I only have some minor considerations:

- Did authors studied vestibule oculomotor reflex in patients?

- Moreover, in some studies the proposed rehab exercises are reported? Are they mainly focused on postural exercises or on visa vestibular interactions?

If the answer to both questions is that they are not reported extensively, I suggest to include a note in the limitations

Author Response

We are thankful for your consideration and suggestion about our study. We tried to improve our work following your indication. In the following part you will find the point-by-point reply.

- Did authors studied vestibule oculomotor reflex in patients?

The authors of the studies investigated the VOR at baseline assessment. No one analysed this reflex on the follow up. As suggested, we added this point to the limitations (line 370-372). Moreover we added this consideration on the Outcome section of our paper (line 202-204)

- Moreover, in some studies the proposed rehab exercises are reported? Are they mainly focused on postural exercises or on visa vestibular interactions?

As suggested, we added report about exercised used in the studies on the Table 2, some author explained accurately the exercise in other paper that we add as reference of our study. There was one study which did not clearly define the exercise and as suggested we added this point at the limitations section (line 368-369). Moreover we added a brief explanation about progression of the Vestibular Rehabilitation used on the Vestibular Rehabilitation section (line185 – 188).
